# Differences in COVID-19-Related Hospitalization, Treatment, Complications, and Death by Race and Ethnicity and Area-Level Measures Among Individuals with Cancer in the ASCO Registry

**DOI:** 10.3390/cancers17050857

**Published:** 2025-03-02

**Authors:** Adiba Ashrafi, Yong Lin, Angela J. Fong, Jessica Y. Islam, Tiffany C. Turner Anderson, Shridar Ganesan, Carolyn J. Heckman, Adana A. M. Llanos

**Affiliations:** 1Department of Epidemiology, Mailman School of Public Health, Columbia University Irving Medical Center, 722 West 168th Street, Room 720G, New York, NY 10032, USA; 2Department of Biostatistics and Epidemiology, Rutgers School of Public Health, Piscataway, NJ 08854, USA; 3Rutgers Cancer Institute, New Brunswick, NJ 08901, USA; 4School of Kinesiology and Rogel Cancer Center, University of Michigan, Ann Arbor, MI 48109, USA; 5Cancer Epidemiology Program, H. Lee Moffitt Cancer Center and Research Institute, Tampa, FL 33612, USA; 6Department of Medicine, Georgetown University Medical Center, Washington, DC 20007, USA; 7Department of Medicine and Pharmacology, Rutgers Robert Wood Johnson Medical School, New Brunswick, NJ 08901, USA; 8Department of Medicine, Rutgers Robert Wood Johnson Medical School, New Brunswick, NJ 08901, USA; 9Herbert Irving Comprehensive Cancer Center, Columbia University Irving Medical Center, New York, NY 10032, USA

**Keywords:** COVID-19, cancer, cancer survivorship, social and structural drivers of health inequities

## Abstract

Individuals with cancer who are exposed to SARS-CoV-2, the virus that causes coronavirus disease 2019 (COVID-19), are more likely to develop COVID-19 complications and die than persons without cancer. During the COVID-19 pandemic, racial and ethnic minorities were more likely than non-Hispanic Whites to contract the virus, be hospitalized, or die. Therefore, this study examines COVID-19-related hospitalization, supplemental oxygen need, multiorgan complications, and death in a large sample of multiethnic cancer patients with SARS-CoV-2 infection from the American Society of Clinical Oncology’s COVID-19 Registry. Cancer patients from racial and ethnic minority groups, as well as those living in socioeconomically disadvantaged areas, were identified to be at a significantly higher risk of poorer COVID-19-related outcomes than their non-Hispanic White counterparts and those living in less disadvantaged areas.

## 1. Introduction

Individuals with cancer exposed to SARS-CoV-2, the virus that causes coronavirus disease 2019 (COVID-19), are more susceptible to COVID-19-related complications [1,2,3,4,5,6,7,8] and death [4,6,7,8,9,10,11,12,13,14,15,16,17,18,19,20] than individuals without cancer due to their underlying disease, pre-existing comorbidities (e.g., cardiovascular disease [CVD], hypertension, diabetes, and obesity), and treatment-related immunosuppression [3,4,5,6,11,19,20,21]. In the general population, compared to non-Hispanic White (NHW) individuals, racial and ethnic minorities—namely, American Indian/Alaska Native (AIAN), non-Hispanic Black (NHB), and Hispanic/Latinx individuals—shouldered a disproportionate burden of COVID-19 infection, hospitalization, and death during the COVID-19 pandemic [21,22,23,24,25,26,27,28]. Extant data have documented differences in COVID-19-related outcomes by individuals’ sociodemographic characteristics and by area-level measures of social and structural drivers of health (SDOHs) [1,4,11,21,29,30,31,32,33], including neighborhood contextual [27,34,35,36,37] and social vulnerability factors [38,39].

In this study, we evaluated COVID-19-related hospitalization, supplemental oxygen need, multiorgan complications, and death (at the time of confirmed SARS-CoV-2 infection [T_0_] and to the end of the acute phase of infection approximately 1–3 months after a positive test [T_1_]) in a large sample of cancer patients with SARS-CoV-2 infection (>5000) from the American Society of Clinical Oncology COVID-19 Registry (ASCO Registry), assessing differences in these outcomes by socially constructed race and ethnicity and area-level SDOH measures, including population density, median household income, and percentage of population with only a high school (HS) diploma, ≤64 years of age without health insurance, reporting White race, and reporting Hispanic/Latinx ethnicity in patients’ area of residence at cancer diagnosis. Our hypothesis was that NHB and Hispanic/Latinx cancer patients, as well as those residing in socioeconomically disadvantaged areas (e.g., characterized by lower median household income, lower educational attainment, higher percentage uninsured, higher proportions of racial and ethnic minority residents, and/or living in a rural area with limited access to healthcare), experience worse COVID-19-related complications and have an increased risk of death, which may contribute to a worsening of cancer inequities in the years to come.

## 2. Materials and Methods

### 2.1. Study Design

This was a retrospective, registry-based cohort study, conducted using data from the ASCO Registry in compliance with the Strengthening the Reporting of Observational Studies in Epidemiology (STROBE) reporting guidelines for cohort studies. As previously described [40,41], starting in April 2020, oncology practices (including private, hospital/health system, and academic practices) participating in the ASCO Registry identified cancer patients with a confirmed SARS-CoV-2 infection. For the current analysis, we received a limited dataset (N = 5262), which included data on cancer patients with a confirmed SARS-CoV-2 infection through July 2022. Our analysis excluded patients of missing/unknown or “other/mixed” race or ethnicity (N = 28), sex (N = 8), age at confirmed SARS-CoV-2 infection (N = 1), and/or vital status (N = 79). Consequently, the analytic sample comprised 5146 individuals with cancer who had available data on COVID-19-related outcomes in the ASCO Registry. Among them, 3314 individuals had short-term follow-up data post the acute phase of infection.

### 2.2. Outcomes

The outcomes of interest were COVID-19-related hospitalization (including admission to the intensive care unit [ICU]), COVID-19-related receipt of supplemental oxygen, the prevalence of any COVID-19-related complications (pneumonia, systemic complications [bleeding, disseminated intravascular coagulation, sepsis, and multiorgan failure]), pulmonary complications (acute respiratory distress syndrome (ARDS), pneumonitis, pulmonary embolism, and respiratory failure), cardiovascular complications [cardiac arrhythmia, cerebrovascular accident, congestive heart failure, deep venous thrombosis, and myocardial infarction], and other complications [acute hepatic injury, bowel perforation, peritonitis, acute renal failure, encephalopathy, and seizures]), and COVID-19-related death. The need for mechanical ventilation due to COVID-19 was also measured but not considered as a primary outcome due to the limited number of indicated patients, resulting in insufficient statistical power to model it as an outcome.

### 2.3. Predictors and Covariates

The main predictor of interest, race and ethnicity (non-Hispanic Asian American and Pacific Islander [AAPI], non-Hispanic American Indian or Alaska Native [AIAN], Hispanic/Latinx [all races], non-Hispanic Black [NHB], and non-Hispanic White [NHW]), was identified using medical record data. Other individual-level covariates included sociodemographic factors (sex and age at COVID-19 diagnosis), COVID-19-related factors (time of COVID-19 diagnosis, vaccination status [as of December 2020], and receipt of anti-COVID-19 drugs), health history factors (body mass index [BMI], history of tobacco use, and number of comorbidities requiring active treatment in the past 12 months), and cancer-related factors (cancer type [breast; gastrointestinal; gynecologic or genitourinary; lymphoid, hematopoietic, or related tissue; respiratory or intrathoracic organ; other], cancer stage [local, regional, metastatic, cancer-free but receiving adjuvant therapy, no solid tumor], receipt of systemic anti-cancer therapy [radiotherapy, pharmacotherapy, transplant/cellular therapy], and last-known performance status using the Eastern Cooperative Oncology Group [ECOG] scale). The number of pulmonary and renal conditions patients had and whether they received COVID-19 or cancer treatment via telemedicine, therapeutic clinical trials, or other treatment approaches were also determined but not associated with any of the outcomes of interest and therefore excluded from multivariable models. Patients’ current cancer status (stable, progressing, responding to treatment, cancer-free but receiving adjuvant therapy, no solid tumor, or unknown) was assessed but excluded due to its correlation with cancer stage.

We further explored the relationship between geographic and area-level SDOH factors measured in the patients’ residential area at cancer diagnosis with the outcomes of interest. Data on these geographic and area-level measures, including census region, population density, median household income, and percentage of population with only a high school (HS) diploma, ≤64 years without health insurance, reporting White race, and reporting Hispanic/Latinx ethnicity, were obtained from the Agency of Healthcare Research and Quality’s Social Determinants of Health Database (specifically, zip code-level data derived from the Census Bureau’s American Community Survey 5-year estimates) [40]. To prevent the possibility of participant reidentification, the area-level SDOH variables were segmented into quartiles (Q1–Q4).

The ASCO Registry also collected data on the census region and population density of oncology practices where registry participants were enrolled (shown in the descriptive analyses), but because these variables were highly correlated with their counterparts based on the patients’ area of residence, they were excluded from the regression analyses. The multivariable models included census region, percentage of population with only an HS diploma, and median household income as area-level measures, since they were significantly associated with the outcomes of interest.

### 2.4. Statistical Analysis

Characteristics of the ASCO Registry patients were compared across racial and ethnic groups using Chi-Squared or Fisher’s exact tests and Analysis of Variance (ANOVA) for categorical and continuous variables, respectively. Independent relative risk (RR) regression models were run using the generalized linear model procedure with Poisson distribution, log link, and robust error variances to examine the association between race and ethnicity or area-level SDOH measures with the primary outcomes (COVID-19-related hospitalization, supplemental oxygen need, complications, and death), at the time of positive SARS-CoV-2 infection (T_0_, N = 5146) and from the time of positive SARS-CoV-2 infection to the end of the acute phase of infection (T_0_ − T_1_, N = 3314). First, associations between race and ethnicity with the outcomes of interest were assessed in models adjusted for individual-level covariates (sociodemographics, COVID-19-related factors, health history, and cancer-related factors), and then area-level covariates (census region, percentage of population with only an HS diploma, and median household income) were added.

The Benjamini–Hochberg (BH) approach was employed to correct for multiple comparisons, using a False Discovery Rate (FDR) of 0.05. Following correction, descriptive analyses with *p*-values ≤0.021 and RR regression analyses with *p*-values ≤ 0.003 were considered statistically significant. SPSS (version 26.0) and STATA (version 16.1) were used for descriptive statistics and regression analyses, respectively.

## 3. Results

### 3.1. Cohort Characteristics

The study sample comprised 66.6% NHW, 13.1% Hispanic/Latinx, 12.3% NHB, 4.2% AAPI, and 3.8% AIAN patients (Table 1). More than half were female (57.0%), and the majority (62.7%) tested positive for SARS-CoV-2 at an older age (≥60 years). Of those diagnosed at younger ages (<60 years), a larger proportion were AIAN, Hispanic/Latinx, and NHB than NHW (*p* < 0.001). Relative to NHWs, racial and ethnic minorities generally resided in areas marked by greater socioeconomic disadvantage (i.e., larger proportions with a lower median household income, only an HS diploma, and no health insurance; *p* < 0.001).

### 3.2. Prevalence of COVID-19-Related Hospitalization, Complications, and Death

At certain time points during the pandemic, significantly higher proportions of NHB (January–April 2020: 15.0%; May–August 2020: 21.6%), Hispanic/Latinx (May–August 2020: 25.4%), and AAPI cancer patients (January–April 2020: 11.5%; May–August 2020: 19.4%) tested positive for SARS-CoV-2 compared to NHWs (January–April 2020: 5.4%; May–August 2020: 12.3%) (Table 2). In addition to a low vaccination rate in the overall study sample (26.5%), fewer NHBs and AIANs were vaccinated than NHWs (17.8% and 18.5% vs. 27.9%, respectively). COVID-19-related hospitalization (without an ICU visit) was more prevalent among NHB than among NHW and Hispanic/Latinx individuals (29.9% vs. 24.5% and 20.2%, respectively), while ICU admission did not differ by race and ethnicity. Fewer Hispanic/Latinx than NHW and NHB patients required supplemental oxygen (15.0% vs. 21.2% and 25.2%) and received anti-COVID-19 medications like Remdesivir, Hydroxychloroquine, Azithromycin Dexamethasone, convalescent plasma, monoclonal antibodies, and others (22.4% vs. 29.7% and 31.8%), whereas more NHBs than NHWs (5.2% vs. 2.6%) required mechanical ventilation.

At the time of a positive SARS-CoV-2 test, NHB patients had a higher prevalence of COVID-19-related complications compared to NHW and Hispanic/Latinx patients (51.1% vs. 39.5% and 39.8%, respectively). COVID-19-related pneumonia was more prevalent in NHB than in NHW and Hispanic/Latinx patients (29.6% vs. 22.9% and 18.4%, respectively). Other COVID-19-related pulmonary complications (i.e., ARDS, pneumonitis, pulmonary embolism, and respiratory failure) did not vary significantly by race and ethnicity (*p* = 0.16). Acute renal failure was more prevalent in NHBs (38.1%) than in other racial and ethnic groups (Appendix A). AAPIs had a higher proportion of COVID-19-related systemic complications than AIAN, Hispanic/Latinx, and NHW patients (16.6% vs. 5.6%, 9.5%, and 7.1%, respectively), and more cardiovascular complications than Hispanic/Latinx patients (8.8% vs. 3.0%, respectively) (Table 2). In total, 260 (5.1%) individuals died of COVID-19 and related complications immediately following SARS-CoV-2 infection. While deaths due to COVID-19 did not differ significantly by race and ethnicity, deaths due to other causes (i.e., cancer, complications of cancer treatment) were twice as common among NHBs than NHWs (4.6% vs. 2.3%). Among the 3314 cohort members with available short-term follow-up data, similar patterns of differences by race and ethnicity were observed, although many did not reach the same level of statistical significance (Appendix A).

### 3.3. COVID-19-Related Outcomes by Race and Ethnicity

Many significant associations were observed between race and ethnicity and COVID-19-related hospitalization, receipt of supplemental oxygen, multiorgan complications, and death among participants in the ASCO Registry, immediately following SARS-CoV-2 infection (Table 3). Adjusting for individual-level covariates, NHBs had a 15% greater risk of COVID-19-related hospitalization (RR, 1.15; 95% CI: 1.05–1.26; *p* = 0.003) and a 16% higher risk of developing COVID-19-related complications (RR, 1.16; 95% CI: 1.07–1.27; *p* = 0.001) than NHWs. Hispanic/Latinx patients had a 21% greater risk of developing COVID-19-related complications (RR, 1.21; 95% CI: 1.09–1.34; *p* < 0.001) compared to NHWs. Although not significant following BH correction for multiple comparisons, AIANs had a 21% higher risk of COVID-19-related hospitalization (RR, 1.21; 95% CI: 1.02–1.42; *p* = 0.024), 30% higher need for supplemental oxygen (RR, 1.30; 95% CI: 1.03–1.64; *p* = 0.029), 23% higher risk of developing COVID-19-related complications (RR, 1.23; 95% CI: 1.04–1.47; *p* = 0.019), and 70% higher risk of death due to COVID-19 or COVID-19-related complications (RR, 1.70; 95% CI: 1.02–2.83; *p* = 0.043) than NHWs.

With an additional adjustment for area-level measures (census region, percentage of population with only a high school diploma, and median household income), racial and ethnic differences persisted: NHB and Hispanic/Latinx patients had a 17% (RR, 1.17; 95% CI: 1.07–1.29; *p* = 0.001) and 20% (RR, 1.20; 95% CI: 1.07–1.34; *p* = 0.001) greater risk, respectively, of COVID-19-related complications at the time of SARS-CoV-2 infection relative to NHWs (Table 3). AAPIs were marginally more likely to be hospitalized due to COVID-19 than NHWs (RR, 1.17; 95% CI: 0.99–1.37; *p* = 0.063). Afterwards, when cohort members were followed from T_0_ to T_1_, Hispanic/Latinx patients had a 29% higher risk of COVID-19-related complications (RR, 1.29; 95% CI: 1.13–1.47; *p* < 0.001) than NHW patients, after controlling for individual- and area-level confounders (Table 4). NHBs were also marginally more likely to be hospitalized through the end of the acute phase of infection than NHWs (RR, 1.21; 95% CI: 0.99–1.47; *p* = 0.066).

### 3.4. COVID-19-Related Outcomes by Area-Level Measures

Generally, residing in areas with higher proportions of the population having only an HS diploma was associated with poorer COVID-19-related outcomes. At the time of SARS-CoV-2 infection, relative to patients residing in Q1 areas (<25.5% with only HS education), those residing in Q2 areas (25.5–33.7% with only HS education) had a 16% higher need for supplemental oxygen (RR, 1.16; 95% CI: 1.02–1.32; *p* = 0.025) (Table 5). Residents in Q3 areas (33.8–41.2% with only HS education) had a 17% greater risk of COVID-19-related hospitalization (RR, 1.17; 95% CI: 1.05–1.30; *p* = 0.005), 24% greater need for supplemental oxygen (RR, 1.24; 95% CI: 1.06–1.44; *p* = 0.006), 15% greater risk of complications (RR, 1.15; 95% CI: 1.03–1.28; *p* = 0.011), and 56% greater risk of death (RR, 1.56; 95% CI: 1.05–2.31; *p* = 0.029). Residents of Q4 areas (≥42.2% with only HS education) had a 29% greater need for supplemental oxygen (RR, 1.29; 95% CI: 1.06–1.57; *p* = 0.011) and 112% greater risk of death (RR, 2.12; 95% CI: 1.33–3.39; *p* = 0.002), the latter being significant following BH correction for multiple comparisons. Also, when followed from time of SARS-CoV-2 infection to the end of the acute phase of disease, cancer patients residing in Q4 (≥42.2% with only HS education) vs. Q1 (<25.5% with only HS education) areas had a 23% increased risk of developing COVID-19-related complications (RR, 1.23; 95% CI: 1.01–1.50; *p* = 0.036).

For median household income, significant differences across the quartiles were not observed for any of the COVID-19-related outcomes, following BH correction for multiple comparisons. This held true with or without adjustment for percentage of population with only a high school diploma. Among the marginal associations observed, we found that residence in areas with the lowest median household annual income (Q1: <USD 43,125) was associated with a 20% lower likelihood of receiving supplemental oxygen at the time of SARS-CoV-2 infection (RR, 0.80; 95% CI: 0.68–0.96; *p* = 0.013) and throughout the short-term follow-up period (RR, 0.80; 95% CI: 0.63–1.01; *p* = 0.057) compared to residence in areas with the highest median household income (Q4: ≥USD 68,446). Also, residence in Q1 vs. Q4 areas of median household income was associated with a 16% lower risk of developing any COVID-19-related complications (RR, 0.84; 95% CI: 0.72–0.99; *p* = 0.033) by the end of short-term follow-up. In contrast, COVID-19-related deaths were more likely to occur during the short-term follow-up period among cancer patients residing in areas with a lower median household income (Q1 [<USD 43,125] vs. Q4 [≥USD 68,446]: RR, 1.63; 95% CI: 0.70–3.77; *p* = 0.254; Q2 [USD 43,125-USD 54,047] vs. Q4 [≥USD 68,446]: RR, 1.96; 95% CI: 1.03–3.71; *p* = 0.039; Q3 [USD 54,048-USD 68,446] vs. Q4 [≥USD 68,446]: RR, 1.71; 95% CI: 0.93–3.16; *p* = 0.087).

## 4. Discussion

These findings support our hypothesis and prior data showing that COVID-19-related outcomes disproportionately affect racial and ethnic minority cancer patients compared with NHWs. NHB cancer patients had a 15% and 17% greater risk of being hospitalized due to COVID-19 and developing COVID-19-related complications, respectively, than NHW cancer patients immediately following SARS-CoV-2 infection. Hispanic/Latinx patients were 20% and 29% more likely to experience COVID-19-related complications at the time of SARS-CoV-2 infection and during the short-term follow-up period, respectively, than NHWs. These findings align with prior evidence showing that NHB and Hispanic/Latinx individuals with COVID-19 were more likely to experience more severe COVID-19 disease, to be hospitalized, admitted to the ICU, and/or experience longer hospital stays than NHWs [4,21,23,24,28,29,30,42,43,44,45,46,47,48,49,50,51,52,53,54,55]. Consistent with a study evaluating SARS-CoV-2 infection among veterans [52], showing that Black racial identity was associated with an increased need for mechanical ventilation, we also observed greater need for mechanical ventilation among NHB cancer patients in the ASCO Registry.

In this study, AAPI cancer patients had a marginally higher risk of hospitalization immediately following SARS-CoV-2 infection, which is consistent with findings from a larger study involving 70,564 AAPIs in California [56]. Among cancer patients, previous studies have shown that the severity of COVID-19-related pneumonia and low oxygen saturation predicted ICU admission [48,57], whereas chemotherapy [12], advanced age, male sex, and kidney failure [4,48,49] predicted COVID-19-related death. In contrast to other study findings showing an increased risk of COVID-19-related death among Black [4,29] and Hispanic cancer patients [29], and conversely, a lower risk of COVID-19-related death among AAPI and multiracial patients [29], we did not observe any significant differences between NHB, Hispanic/Latinx, or AAPI race and ethnicity and COVID-19-related death. Instead, we observed a marginally higher risk of COVID-19-related death among AIANs in our cohort. This finding supports a recent analysis from the COVID-19 Critical Care Consortium [26] where the investigators reported high rates of mechanical ventilation and COVID-19-related mortality among AIANs in the US.

Findings from the current study additionally showed that NHB, Hispanic/Latinx, and AIAN cancer patients have an increased risk of developing COVID-19-related complications. Due to the distribution of upstream factors (social disadvantage, risk exposure, and social inequities in education, employment, housing, and access to resources), many racial and ethnic minority individuals were more vulnerable to viral infection [21,46,58,59] and food/supply shortages [60] and were less likely to have received a vaccination to help mitigate COVID-19’s severity [61] and were less likely to receive timely, high-quality medical care during the pandemic [21,45,60]. All these factors contribute to elevated risks of more severe acute illness and long-term sequelae following infection with SARS-CoV-2 [62,63]. Furthermore, these vulnerable groups tend to disproportionately reside in social environments and communities with high concentrations of unhealthy food, tobacco, and alcohol (i.e., food deserts, food swamps, areas with high densities of alcohol and tobacco retailers) [64,65,66], further increasing the risk of developing chronic conditions (i.e., diabetes, CVD, obesity) that compromise immunity and contribute to worse COVID-19 outcomes [21,47,59,61,67].

Our study is among those that have explored the relationship between area-level measures and COVID-19-related outcomes [27,28,37,55,68,69,70,71,72,73]. We found that cancer patients living in areas with a higher percentage of individuals who only possess an HS diploma were at greater risk for COVID-19-related hospitalization, had an increased need for supplemental oxygen, and were more likely to experience complications or death due to COVID-19. Additionally, cancer patients who lived in areas characterized by a lower median household income were more likely to die from COVID-19 or related complications. Other area-level measures, such as population density, percentage uninsured, and percentage White, were not significantly associated with COVID-19 outcomes among cancer patients in the ASCO Registry, which contrasts with findings from other studies showing that residence in census tracts with marked crowding, areas with a high percentage of uninsured adults, a high unemployment rate, and Black-concentrated neighborhoods increased COVID-19-related hospitalization and mortality [27,28,54,55,70,74]. This could be due to the limited granularity of the area-level measures available in the current study.

While several prior studies have evaluated COVID-19-related disparities in hospitalization, treatment, complications, and death [1,4,11,21,27,28,29,30,31,32,37,54,69,75], ours appears to be the first longitudinal analysis of COVID-19 outcomes in cancer patients infected between mid-2020 and mid-2022, with some cohort participants having follow-up data available after the acute COVID-19 disease phase. Furthermore, this study is unique in that it examined differences in the receipt of supplemental oxygen and the development of multiorgan complications. This study had a large overall sample size of >5000 patients, although the sizes of the AAPI and AIAN subgroups were quite small, reducing the statistical power to detect meaningful associations in these subgroups. Additional limitations that should be considered include the consideration that the use of a registry-based sample of individuals with cancer who sought testing for SARS-CoV-2 infection may introduce selection bias. Some individuals may not have pursued laboratory-based SARS-CoV-2 testing for various reasons, while others may have received false-negative results, rendering them ineligible for inclusion in this study. Further, individuals with cancer included in the ASCO Registry represent those actively engaged in oncology care and who had survived long enough to receive cancer treatment, which may limit the generalizability of our findings to the broader cancer population. Another point to consider is that the reliance on passive data collection, particularly for race and ethnicity, rather than self-reporting, may contribute to classification bias. This approach also increases the likelihood of missing data, including on potential confounders of the associations under investigation. This is especially relevant when examining the relationships between area-level SDOHs and the outcomes of interest, where missing data may impact the robustness of our conclusions. Lastly, while longitudinal data were obtained following confirmed COVID-19 diagnosis in the ASCO Registry, the availability of each clinical data point and loss to follow-up varied, resulting in differential missingness, which also impacts our findings.

## 5. Conclusions

Despite these limitations, our findings demonstrate that cancer patients from racial and ethnic minority groups, as well as those living in socioeconomically disadvantaged areas, face a significantly greater risk of poorer COVID-19-related outcomes than their NHW counterparts and those in less disadvantaged areas. These results provide critical insights for oncology clinicians, highlighting the need to consider differences in COVID-19 sequelae when delivering optimal care—particularly for marginalized patient populations that already experience a higher cancer mortality burden and have also shouldered a greater burden from COVID-19. To address these inequities, policy-based solutions are needed, including the allocation of additional resources to communities with high proportions of Black and/or Hispanic cancer patients with lower socioeconomic status, as well as to the healthcare facilities that serve these populations. Targeted investments could enhance access to basic health needs, such as nutritious food, stable housing, reliable transportation, and comprehensive healthcare services such as COVID-19 vaccination and clinicians equipped to manage COVID-19-related complications. At the clinic and hospital level, expanding community health programs and patient navigation programs represents a promising strategy to reduce inequities in care. These programs can leverage electronic health records to identify patients in need of additional support and provide continuity of care beyond hospital discharge. By assisting with appointment scheduling, medication access, and adherence to preventive health recommendations, these initiatives can help bridge critical gaps in care and improve health outcomes. Ultimately, this study confirms the presence of structural issues that must be addressed to advance health equity and social justice. As future infectious disease outbreaks are likely to disproportionately affect marginalized oncology patients, implementing both policy-based and healthcare system-level interventions will be essential to mitigating disparities and ensuring equitable cancer care.

## Figures and Tables

**Table 1 cancers-17-00857-t001:** Characteristics of cancer patients in the ASCO Registry at time of confirmed SARS-CoV-2 infection, overall and by race and ethnicity, N = 5146.

PATIENT CHARACTERISTICS	AT CONFIRMED SARS-CoV-2 INFECTION (T_0_) ^‡^						
	Overall N (%)	AAPI N (%)	AIAN N (%)	Hispanic N (%)	NHB N (%)	NHW N (%)	*p* ^†^
TOTAL	5146 (100.0)	217 (4.2)	195 (3.8)	673 (13.1)	635 (12.3)	3426 (66.6)	
SOCIODEMOGRAPHICS							
Sex							0.13
Male	2212 (43.0)	90 (41.5)	89 (45.6)	277 (41.2)	248 (39.1)	1508 (44.0)	
Female	2934 (57.0)	127 (58.5)	106 (54.4)	396 (58.8)	387 (60.9)	1918 (56.0)	
Age at COVID-19 diagnosis (years)							<0.001
<50	887 (17.2)	50 (23.0)	36 (18.5)	209 (31.1)	145 (22.8)	447 (13.0)	
50–59	1030 (20.0)	45 (20.7)	49 (25.1)	181 (26.9)	126 (19.8)	629 (18.4)	
60–69	1446 (28.1)	49 (22.6)	49 (25.1)	157 (23.3)	186 (29.3)	1005 (29.3)	
≥70	1783 (34.6)	73 (33.6)	61 (31.3)	126 (18.7)	178 (28.0)	1345 (39.3)	
HEALTH HISTORY							
Body mass index (kg/m^2^) ^a^							0.005
<25.00	1467 (28.5)	81 (37.3)	57 (29.2)	173 (25.7)	175 (27.6)	981 (28.6)	
25.00–29.99	1585 (30.8)	66 (30.4)	69 (35.4)	204 (30.3)	169 (26.6)	1077 (31.4)	
30.00–34.99	1032 (20.1)	31 (14.3)	39 (20.0)	145 (21.5)	130 (20.5)	687 (20.1)	
≥35.00	929 (18.1)	32 (14.7)	26 (13.3)	125 (18.6)	142 (22.4)	604 (17.6)	
History of tobacco use							<0.001
Never smoker	2540 (49.4)	103 (47.5)	101 (51.8)	434 (64.5)	313 (49.3)	1589 (46.4)	
Former smoker	1975 (38.4)	82 (37.8)	54 (27.7)	185 (27.5)	224 (35.3)	1430 (41.7)	
Current smoker	448 (8.7)	15 (6.9)	15 (7.7)	35 (5.2)	73 (11.5)	310 (9.0)	
Unsure	183 (3.6)	17 (7.8)	25 (12.8)	19 (2.8)	25 (3.9)	97 (2.8)	
Number of comorbidities requiring active treatment in past 12 months							<0.001
0	1908 (37.1)	86 (39.6)	101 (51.8)	320 (47.5)	144 (22.7)	1257 (36.7)	
1	1533 (29.8)	61 (28.1)	52 (26.7)	173 (25.7)	208 (32.8)	1039 (30.3)	
2	995 (19.3)	42 (19.4)	19 (9.7)	120 (17.8)	161 (25.4)	653 (19.1)	
≥3	710 (13.8)	28 (12.9)	23 (11.8)	60 (8.9)	122 (19.2)	477 (13.9)	
Number of pulmonary conditions							<0.001
0	4371 (84.9)	176 (81.1)	178 (91.3)	585 (86.9)	526 (82.8)	2906 (84.8)	
≥1	687 (13.4)	25 (11.5)	14 (7.2)	47 (7.0)	105 (16.5)	496 (14.5)	
Number of renal conditions							<0.001
0	4588 (89.2)	183 (84.3)	183 (93.8)	588 (87.4)	543 (85.5)	3091 (90.2)	
≥1	450 (8.7)	16 (7.4)	11 (5.6)	41 (6.1)	82 (12.9)	300 (8.8)	
CANCER HISTORY							
Cancer type ^b^							<0.001
Breast	1199 (23.3)	47 (21.7)	42 (21.5)	169 (25.1)	157 (24.7)	784 (22.9)	
Gastrointestinal	784 (15.2)	38 (17.5)	32 (16.4)	157 (23.3)	90 (14.2)	467 (13.6)	
Gynecologic or genitourinary	735 (14.3)	35 (16.1)	29 (14.9)	100 (14.9)	108 (17.0)	463 (13.5)	
Lymphoid, hematopoietic, or related tissue	1208 (23.5)	44 (20.3)	40 (20.5)	125 (18.6)	135 (21.3)	864 (25.2)	
Respiratory or intrathoracic organ	597 (11.6)	27 (12.4)	28 (14.4)	25 (3.7)	79 (12.4)	438 (12.8)	
Other	623 (12.1)	26 (12.0)	24 (12.3)	97 (14.4)	66 (10.4)	410 (12.0)	
Cancer stage							0.002
Local	1197 (23.3)	46 (21.2)	51 (26.2)	205 (30.5)	148 (23.3)	747 (21.8)	
Regional	522 (10.1)	25 (11.5)	11 (5.6)	61 (9.1)	70 (11.0)	355 (10.4)	
Metastatic	1781 (34.6)	77 (35.5)	83 (42.6)	211 (31.4)	213 (33.5)	1197 (34.9)	
Cancer-free but receiving adjuvant therapy	280 (5.4)	13 (6.0)	9 (4.6)	33 (4.9)	32 (5.0)	193 (5.6)	
No solid tumor	1366 (26.5)	56 (25.8)	41 (21.0)	163 (24.2)	172 (27.1)	934 (27.3)	
Current cancer status							0.003
Stable	1510 (29.3)	49 (22.6)	52 (26.7)	220 (32.7)	177 (27.9)	1012 (29.5)	
Progressing	724 (14.1)	33 (15.2)	33 (16.9)	102 (15.2)	97 (15.3)	459 (13.4)	
Responding to treatment	313 (6.1)	11 (5.1)	4 (2.1)	42 (6.2)	39 (6.1)	217 (6.3)	
Cancer-free but receiving adjuvant therapy	280 (5.4)	13 (6.0)	9 (4.6)	33 (4.9)	32 (5.0)	193 (5.6)	
No solid tumor	1366 (26.5)	56 (25.8)	41 (21.0)	163 (24.2)	172 (27.1)	934 (27.3)	
Unknown	939 (18.2)	55 (25.3)	56 (28.7)	112 (16.6)	115 (18.1)	601 (17.5)	
Receipt of systemic anti-cancer therapy ^c^							<0.001
No	1405 (27.3)	64 (29.5)	66 (33.8)	244 (36.3)	179 (28.2)	852 (24.9)	
Yes	3741 (72.7)	153 (70.5)	129 (66.2)	429 (63.7)	456 (71.8)	2574 (75.1)	
Last known performance status (ECOG) ^d^							<0.001
0	1561 (30.3)	58 (26.7)	67 (34.4)	214 (31.8)	149 (23.5)	1073 (31.3)	
1	1505 (29.2)	66 (30.4)	44 (22.6)	176 (26.2)	190 (29.9)	1029 (30.0)	
≥2	665 (12.9)	31 (14.3)	21 (10.8)	58 (8.6)	100 (15.7)	455 (13.3)	
Unknown	1122 (21.8)	50 (23.0)	48 (24.6)	192 (28.5)	148 (23.3)	684 (20.0)	
AREA-LEVEL MEASURES							
Census region of treatment oncology practice							<0.001
West	436 (8.5)	54 (24.9)	11 (5.6)	95 (14.1)	20 (3.1)	256 (7.5)	
Midwest	1554 (30.2)	57 (26.3)	53 (27.2)	70 (10.4)	98 (15.4)	1276 (37.2)	
Northeast	724 (14.1)	30 (13.8)	19 (9.7)	74 (11.0)	115 (18.1)	486 (14.2)	
South	2427 (47.2)	76 (35.0)	110 (56.4)	434 (64.5)	402 (63.3)	1405 (41.0)	
Population density of treatment oncology practice							<0.001
Urban	4866 (94.6)	214 (98.6)	177 (90.8)	661 (98.2)	613 (96.5)	3201 (93.4)	
Rural	275 (5.3)	3 (1.4)	16 (8.2)	12 (1.8)	22 (3.5)	222 (6.5)	
Census region of patient’s primary residence ^e^							<0.001
West	444 (8.6)	54 (24.9)	11 (5.6)	95 (14.1)	20 (3.1)	264 (7.7)	
Midwest	1540 (29.9)	58 (26.7)	53 (27.2)	70 (10.4)	97 (15.3)	1262 (36.8)	
Northeast	719 (14.0)	30 (13.8)	20 (10.3)	74 (11.0)	115 (18.1)	480 (14.0)	
South	2441 (47.4)	75 (34.6)	111 (56.9)	434 (64.5)	402 (63.3)	1419 (41.4)	
Population density of patient’s primary residence ^e^							<0.001
Urban	4416 (85.8)	197 (90.8)	172 (88.2)	627 (93.2)	582 (91.7)	2838 (82.8)	
Rural	728 (14.1)	20 (9.2)	23 (11.8)	46 (6.8)	52 (8.2)	587 (17.1)	
Median household income ^e,f^							<0.001
Q1: <USD 43,125	880 (17.1)	30 (13.8)	22 (11.3)	232 (34.5)	238 (37.5)	358 (10.4)	
Q2: USD 43,125–USD 54,047	1128 (21.9)	37 (17.1)	34 (17.4)	148 (22.0)	140 (22.0)	769 (22.4)	
Q3: USD 54,048–USD 68,446	1263 (24.5)	54 (24.9)	43 (22.1)	128 (19.0)	103 (16.2)	935 (27.3)	
Q4: ≥USD 68,446	1533 (29.8)	85 (39.2)	90 (46.2)	147 (21.8)	133 (20.9)	1078 (31.5)	
Percentage of population with only a high school diploma ^e,f^							<0.001
Q1: <25.5%	1598 (31.1)	99 (45.6)	96 (49.2)	206 (30.6)	201 (31.7)	996 (29.1)	
Q2: 25.5–33.7%	1734 (33.7)	66 (30.4)	49 (25.1)	311 (46.2)	219 (34.5)	1089 (31.8)	
Q3: 33.8–41.2%	1166 (22.7)	35 (16.1)	38 (19.5)	126 (18.7)	171 (26.9)	796 (23.2)	
Q4: ≥42.2%	311 (6.0)	6 (2.8)	6 (3.1)	13 (1.9)	23 (3.6)	263 (7.7)	
Percentage of population (≤64 years) with no health insurance ^e,f^							<0.001
Q1: <4.8%	708 (13.8)	32 (14.7)	37 (19.0)	42 (6.2)	33 (5.2)	564 (16.5)	
Q2: 4.8–8.8%	1446 (28.1)	72 (33.2)	51 (26.2)	103 (15.3)	175 (27.6)	1045 (30.5)	
Q3: 8.9–14.7%	1568 (30.5)	47 (21.7)	63 (32.3)	178 (26.4)	244 (38.4)	1036 (30.2)	
Q4: ≥14.8%	1087 (21.1)	55 (25.3)	38 (19.5)	333 (49.5)	162 (25.5)	499 (14.6)	
Percentage of population reporting White race ^e,f^							<0.001
Q1: <77.3%	1942 (37.7)	125 (57.6)	101 (51.8)	298 (44.3)	516 (81.3)	902 (26.3)	
Q2: 77.4–92.1%	1968 (38.2)	66 (30.4)	59 (30.3)	323 (48.0)	89 (14.0)	1431 (41.8)	
Q3: 92.2–97.4%	729 (14.2)	14 (6.5)	27 (13.8)	34 (5.1)	7 (1.1)	647 (18.9)	
Q4: ≥97.5%	158 (3.1)	1 (0.5)	2 (1.0)	1 (0.1)	2 (0.3)	152 (4.4)	
Percentage of population reporting Hispanic ethnicity ^e,f^							<0.001
Q1: <0.7%	119 (2.3)	1 (0.5)	3 (1.5)	0 (0.0)	11 (1.7)	104 (3.0)	
Q2: 0.7–3.1%	959 (18.6)	18 (8.3)	18 (9.2)	20 (3.0)	134 (21.1)	769 (22.4)	
Q3: 3.2–9.5%	1900 (36.9)	72 (33.2)	67 (34.4)	77 (11.4)	263 (41.4)	1421 (41.5)	
Q4: ≥9.5%	1831 (35.6)	115 (53.0)	101 (51.8)	559 (83.1)	206 (32.4)	850 (24.8)	

Abbreviations: AAPI, non-Hispanic Asian American and Pacific Islander; AIAN, non-Hispanic American Indian or Alaska Native; ECOG, Eastern Cooperative Oncology Group; NHB, non-Hispanic Black; NHW, non-Hispanic White; *p*, *p*-value; Q1–Q4, quartiles 1–4. ^†^ For all categorical variables, comparisons of proportions across race and ethnicity groups were assessed using the two-sided Chi-Squared test. The Benjamini–Hochberg (BH) approach was employed to correct for multiple comparisons, using a False Discovery Rate (FDR) of 0.05. Following correction, *p*-values ≤ 0.021 were statistically significant. ^‡^ Missing data for the overall study sample at entry into the cohort were >2% for BMI (2.6%), number of renal conditions (2.1%), ECOG (5.7%), median household income (6.6%), and percent of population with only a high school diploma (6.5%), with no health insurance (6.5%), reporting White race (6.8%), and reporting Hispanic ethnicity (6.5%). ^a^ For BMI, the following two sets of groups were combined: (a) <18.50 kg/m^2^ (underweight, N = 124) and 18.50–25.00 kg/m^2^ (normal weight, N = 1343) into <25.00 kg/m^2^, and (b) 35.00–39.99 kg/m^2^ (obesity class II, N = 541) and ≥40.00 kg/m^2^ (obesity class III, N = 388) into ≥35.00 kg/m^2^. ^b^ The following are some cancers (not all) that fall into the specified categories: “gastrointestinal” cancers include esophageal, stomach, liver, colorectal, and pancreatic cancers; “gynecologic” cancers are cancers of the female and male reproductive organs; “genitourinary” cancers include prostate, kidney, and bladder cancers; “lymphoid, hematopoietic, and related tissue” cancers include lymphomas and leukemias; “respiratory and intrathoracic organ” cancers include lung, bronchus, and heart cancers; and the “other” cancers include those of the lip, oral cavity, pharynx, bone and articular cartilage, mesothelial and soft tissue, malignant skin, eye, brain, and other parts of the CNS, thyroid and other endocrine glands, and unspecified cancers. ^c^ Systemic anti-cancer therapies include radiation therapy, pharmacotherapy, and transplant or cellular therapy. ^d^ For ECOG status, the value “0” signifies fully active, “1” signifies restricted in physically strenuous activity but ambulatory and able to carry out work of a light or sedentary nature, and “≥2” signifies those who are either capable of only limited self-care or completely disabled. ^e^ Social and structural drivers of health (SDOHs) were estimated based on census-level data for patient’s primary area of residence at cancer diagnosis. ^f^ To prevent reidentification of registry patients by way of their residential area, SDOH factors were segmented into quartiles as follows: median household income (USD 43,125, USD 43,125–54,047, USD 54,048–68,446, ≥USD 68,446), % of population with only a high school [HS] diploma (<25.5%, 25.5–33.7%, 33.8–41.2%, ≥42.2%), % of population ≤64 years without health insurance (<4.8%, 4.8–8.8%, 8.9–14.7%, >14.8%), % of population reporting White race (<77.3%, 77.4–92.1%, 92.2–97.4%, ≥97.5%), % of population reporting Hispanic ethnicity (<0.7%, 0.7–3.1%, 3.2–9.5%, 9.5%).

**Table 2 cancers-17-00857-t002:** COVID-19-related diagnosis, treatment, complications, and death among cancer patients in the ASCO Registry at time of confirmed SARS-CoV-2 infection, overall and by race and ethnicity, N = 5146.

COVID-19 CHARACTERISTICS	AT CONFIRMED SARS-CoV-2 INFECTION (T_0_) ^‡^						
	Overall N (%)	AAPI N (%)	AIAN N (%)	Hispanic N (%)	NHB N (%)	NHW N (%)	*p* ^†^
TOTAL	5146 (100.0)	217 (4.2)	195 (3.8)	673 (13.1)	635 (12.3)	3426 (66.6)	
Time of COVID-19 diagnosis							<0.001
January–April 2020	355 (6.9)	25 (11.5)	18 (9.2)	31 (4.6)	95 (15.0)	186 (5.4)	
May–August 2020	804 (15.6)	42 (19.4)	34 (17.4)	171 (25.4)	137 (21.6)	420 (12.3)	
September–December 2020	1513 (29.4)	61 (28.1)	53 (27.2)	145 (21.5)	149 (23.5)	1105 (32.3)	
January–April 2021	760 (14.8)	28 (12.9)	37 (19.0)	104 (15.5)	97 (15.3)	494 (14.4)	
May–August 2021	373 (7.2)	13 (6.0)	13 (6.7)	65 (9.7)	36 (5.7)	246 (7.2)	
September–December 2021	463 (9.0)	16 (7.4)	16 (8.2)	45 (6.7)	58 (9.1)	328 (9.6)	
January–April 2022	677 (13.2)	21 (9.7)	19 (9.7)	102 (15.2)	49 (7.7)	486 (14.2)	
May–August 2022	201 (3.9)	11 (5.1)	5 (2.6)	10 (1.5)	14 (2.2)	161 (4.7)	
Vaccinated							<0.001
No	1330 (25.8)	48 (22.1)	43 (22.1)	206 (30.6)	174 (27.4)	859 (25.1)	
Yes	1362 (26.5)	54 (24.9)	36 (18.5)	202 (30.0)	113 (17.8)	957 (27.9)	
Unsure	860 (16.7)	42 (19.4)	54 (27.7)	74 (11.0)	92 (14.5)	598 (17.5)	
Patient received any care or treatment (for COVID-19 or cancer) via telemedicine							<0.001
No	2823 (54.9)	94 (43.3)	102 (52.3)	293 (43.5)	368 (58.0)	1966 (57.4)	
Yes	1188 (23.1)	45 (20.7)	50 (25.6)	229 (34.0)	149 (23.5)	715 (20.9)	
Unsure	734 (14.3)	43 (19.8)	32 (16.4)	66 (9.8)	60 (9.4)	533 (15.6)	
Patient received COVID-19 treatment as part of therapeutic clinical trial							<0.001
No	4630 (90.0)	173 (79.7)	179 (91.8)	609 (90.5)	561 (88.3)	3108 (90.7)	
Yes	48 (0.9)	5 (2.3)	1 (0.5)	3 (0.4)	15 (2.4)	24 (0.7)	
COVID-19 TREATMENTS							
Hospitalization							<0.001
No	2389 (46.4)	88 (40.6)	84 (43.1)	323 (48.0)	248 (39.1)	1646 (48.0)	
Yes, but not in ICU	1280 (24.9)	68 (31.3)	46 (23.6)	136 (20.2)	190 (29.9)	840 (24.5)	
Length of hospitalization (days), median (IQR)	5 (3–10)	5 (3–11)	7 (4–13)	6 (3–11)	6 (3–12)	5 (3–10)	0.31
Yes, in ICU	325 (6.3)	15 (6.9)	17 (8.7)	39 (5.8)	48 (7.6)	206 (6.0)	
Length of ICU stay (days), median (IQR)	6 (3–11)	5 (2–12)	5 (3–9)	8 (6–12)	7 (3–15)	5 (2–11)	0.46
Receipt of supplemental oxygen							<0.001
No	2754 (53.5)	111 (51.2)	93 (47.7)	378 (56.2)	333 (52.4)	1839 (53.7)	
Yes	1071 (20.8)	44 (20.3)	40 (20.5)	101 (15.0)	160 (25.2)	726 (21.2)	
Length of treatment (days), median (IQR)	5 (3–10)	6 (2–8)	6 (3–10)	4 (3–10)	5 (3–10)	5 (3–10)	0.73
Unsure or unknown	422 (8.2)	31 (14.3)	23 (11.8)	42 (6.2)	32 (5.0)	294 (8.6)	
Receipt of mechanical ventilation							<0.001
No	3697 (71.8)	149 (68.7)	126 (64.6)	450 (66.9)	467 (73.5)	2505 (73.1)	
Yes	162 (3.1)	7 (3.2)	6 (3.1)	28 (4.2)	33 (5.2)	88 (2.6)	
Length of treatment (days), median (IQR)	6 (2–12)	10 (2–13)	4 (3–7)	14 (8–18)	7 (4–13)	5 (2–11)	0.10
Unsure or unknown	366 (7.1)	27 (12.4)	23 (11.8)	35 (5.2)	22 (3.5)	259 (7.6)	
Anti-COVID-19 drugs							<0.001
No	2308 (44.9)	82 (37.8)	92 (47.2)	325 (48.3)	287 (45.2)	1522 (44.4)	
Yes	1487 (28.9)	75 (34.6)	40 (20.5)	151 (22.4)	202 (31.8)	1019 (29.7)	
Unsure or unknown	451 (8.8)	29 (13.4)	24 (12.3)	44 (6.5)	36 (5.7)	318 (9.3)	
Other treatment approaches							<0.001
No	3197 (62.1)	118 (54.4)	115 (59.0)	388 (57.7)	405 (63.8)	2171 (63.4)	
Yes	337 (6.5)	10 (4.6)	10 (5.1)	30 (4.5)	55 (8.7)	232 (6.8)	
Unsure or unknown	591 (11.5)	38 (17.5)	28 (14.4)	52 (7.7)	54 (8.5)	419 (12.2)	
COVID-19-RELATED COMPLICATIONS or DEATH							
Any complications							<0.001
No	2079 (40.4)	71 (50.0)	72 (56.7)	248 (60.2)	218 (48.9)	1470 (60.5)	
Yes	1479 (28.7)	71 (50.0)	55 (43.3)	164 (39.8)	228 (51.1)	961 (39.5)	
Pneumonia							<0.001
No	2629 (51.1)	93 (42.9)	93 (47.7)	324 (48.1)	290 (45.7)	1829 (53.4)	
Yes	1196 (23.2)	54 (24.9)	46 (23.6)	124 (18.4)	188 (29.6)	784 (22.9)	
Systemic complications ^a^							<0.001
No	3864 (75.1)	126 (58.1)	143 (73.3)	461 (68.5)	466 (73.4)	2668 (77.9)	
Yes	418 (8.1)	36 (16.6)	11 (5.6)	64 (9.5)	64 (10.1)	243 (7.1)	
Pulmonary complications ^a^							0.16
No	3531 (68.6)	123 (56.7)	134 (68.7)	449 (66.7)	418 (65.8)	2407 (70.3)	
Yes	798 (15.5)	37 (17.1)	25 (12.8)	87 (12.9)	109 (17.2)	540 (15.8)	
Cardiovascular complications ^a^							0.002
No	4015 (78.0)	145 (66.8)	153 (78.5)	498 (74.0)	489 (77.0)	2730 (79.7)	
Yes	248 (4.8)	19 (8.8)	6 (3.1)	20 (3.0)	38 (6.0)	165 (4.8)	
Other complications ^a^							<0.001
No	3901 (75.8)	137 (63.1)	151 (77.4)	474 (70.4)	447 (70.4)	2692 (78.6)	
Yes	273 (5.3)	20 (9.2)	7 (3.6)	28 (4.2)	68 (10.7)	150 (4.4)	
Death							0.02
No, living	4748 (92.3)	196 (90.3)	177 (90.8)	632 (93.9)	569 (89.6)	3174 (92.6)	
No, death due to cancer, complication of cancer treatment, other or unknown cause	137 (2.7)	7 (3.2)	4 (2.1)	18 (2.7)	29 (4.6)	79 (2.3)	
Yes, COVID-19-related death	260 (5.1)	14 (6.5)	14 (7.2)	23 (3.4)	37 (5.8)	172 (5.0)	
Length of time from diagnosis to death (days), median (IQR)	14 (6–33)	10 (5–20)	15 (8–21)	18 (7–45)	16 (6–33)	13 (6–32)	0.64

Abbreviations: AAPI, non-Hispanic Asian American and Pacific Islander; AIAN, non-Hispanic American Indian or Alaska Native; ICU, intensive care unit; IQR, interquartile range; NHB, non-Hispanic Black; NHW, non-Hispanic White; *p*, *p*-value. ^†^ For all categorical variables, comparisons of proportions across race and ethnicity groups were assessed using the two-sided Chi-Squared test. For continuous variables, comparisons of means across race and ethnicity groups were assessed using Analysis of Variance (ANOVA). The Benjamini–Hochberg (BH) approach was employed to correct for multiple comparisons, using a False Discovery Rate (FDR) of 0.05. Following correction, *p*-values ≤ 0.021 were statistically significant. ^‡^ At time of confirmed SARS-CoV-2 infection, missing data for the overall study sample were >2% for the following categorical variables: vaccination status (31.0%); patient received any care or treatment (for COVID-19 or cancer) via telemedicine (7.8%); patient received treatment for COVID-19 as part of therapeutic clinical trial (9.1%); receipt of COVID-19 treatments such as hospitalization (22.4%), supplemental oxygen (17.5%), mechanical ventilation (17.9%), anti-COVID-19 drugs (17.5%), and other treatment approaches (19.8%); any COVID-19-related complications (30.9%), including pneumonia (25.7%), systemic complications (16.8%), pulmonary complications (15.9%), cardiovascular complications (17.2%), and other complications (18.9%). For continuous variables, the percentages of missing data were as follows: days on supplemental oxygen (58.6%), days on mechanical ventilation (40.1%), days hospitalized but not in ICU (26.7%), days hospitalized and in ICU (44.0%), days from diagnosis to COVID-19-related death (0.5%). ^a^ Systemic complications include bleeding, disseminated intravascular coagulation, and sepsis; pulmonary complications include acute respiratory distress syndrome, pneumonitis, pulmonary embolism, and respiratory failure; cardiovascular complications include cardiac arrhythmia, cerebrovascular accident, congestive heart failure, deep venous thrombosis, and myocardial infarction; and other complications include acute hepatic injury, bowel perforation, peritonitis, acute renal failure, encephalopathy, and seizures.

**Table 3 cancers-17-00857-t003:** Association between socially constructed race and ethnicity and COVID-19-related outcomes among cancer patients in the ASCO Registry at time of confirmed SARS-CoV-2 infection.

Race and Ethnicity	AT CONFIRMED SARS-CoV-2 INFECTION ^†^							
	COVID-19-Related Hospitalization		COVID-19-Related Receipt of Supplemental Oxygen		Any COVID-19-Related Complications ^a^		COVID-19-Related Death ^b^	
	RR (95% CI), *p*		RR (95% CI), *p*		RR (95% CI), *p*		RR (95% CI), *p*	
	Model 1 ^††^	Model 2 ^‡^	Model 1 ^††^	Model 2 ^‡^	Model 1 ^††^	Model 2 ^‡^	Model 1 ^††^	Model 2 ^‡^
N	N = 3893	N = 3637	N = 3733	N = 3487	N = 3478	N = 3245	N = 4878	N = 4553
AAPI	1.10 (0.93–1.28), 0.263	1.17 (0.99–1.37), 0.063	0.96 (0.76–1.22), 0.738	1.02 (0.80–1.29), 0.880	1.10 (0.95–1.28), 0.192	1.12 (0.96–1.30), 0.153	1.09 (0.65–1.83), 0.731	1.20 (0.70–2.05), 0.503
AIAN	1.21 (1.02–1.42), 0.024	1.22 (1.03–1.45), 0.022	1.30 (1.03–1.64), 0.029	1.30 (1.01–1.67), 0.043	1.23 (1.04–1.47), 0.019	1.24 (1.04–1.50), 0.019	1.70 (1.02–2.83), 0.043	1.74 (1.02–2.98), 0.043
Hispanic	1.09 (0.98–1.22), 0.109	1.13 (1.00–1.26), 0.045	1.10 (0.94–1.28), 0.221	1.13 (0.96–1.33), 0.148	**1.21** **(1.09–1.34),** **<0.001**	**1.20** **(1.07–1.34),** **0.001**	1.00 (0.67–1.50), 0.999	1.19 (0.79–1.80), 0.409
NHB	**1.15** **(1.05–1.26),** **0.003**	1.15 (1.04–1.27), 0.005	1.01 (0.89–1.14), 0.873	1.05 (0.92–1.19), 0.494	**1.16** **(1.07–1.27),** **0.001**	**1.17** **(1.07–1.29),** **0.001**	1.01 (0.73–1.38), 0.972	1.08 (0.76–1.53), 0.671
NHW	1.00 (ref.)	1.00 (ref.)	1.00 (ref.)	1.00 (ref.)	1.00 (ref.)	1.00 (ref.)	1.00 (ref.)	1.00 (ref.)

Abbreviations: AAPI, non-Hispanic Asian American and Pacific Islander; AIAN, non-Hispanic American Indian or Alaska Native; NHB, non-Hispanic Black; NHW, non-Hispanic White; *p*, *p*-value; RR, Risk Ratio. ^†^ Independent relative risk regression models were run using a Poisson distribution, log link, and robust error variances to obtain RRs that examined the association between race and ethnicity and COVID-19-related hospitalization, COVID-19-related receipt of supplemental oxygen, any COVID-19-related complications, and COVID-19-related death among cancer patients immediately following SARS-CoV-2 infection (T_0_). The Benjamini–Hochberg (BH) approach was employed to correct for multiple comparisons, using a False Discovery Rate (FDR) of 0.05. Following correction, *p*-values ≤ 0.003 were statistically significant—each significant RR and *p*-value following correction is bolded. ^††^ Model 1 adjusts patient-level covariates–that is, sociodemographic, COVID-19, health history, and cancer status factors measured at time of confirmed SARS-CoV-2 infection. Sociodemographic factors include sex and age at COVID-19 diagnosis; COVID-19 factors include time of COVID-19 diagnosis, vaccination status, and receipt of anti-COVID-19 drugs; health history factors include body mass index, history of tobacco use, and number of comorbidities requiring active treatment in the past 12 months; and cancer status factors include cancer type, cancer stage, receipt of systemic anti-cancer therapy, and last known performance status (ECOG). ^‡^ Model 2 adjusts for all patient-level covariates in Model 1 in addition to area-level measures estimated in the patient’s primary area of residence at the time of cancer diagnosis: census region, percentage of population with only a high school diploma, and median household income. ^a^ Any COVID-19-related complications include the following: COVID-19-related pneumonia, COVID-19-related systemic complications such as bleeding, disseminated intravascular coagulation, and sepsis; COVID-19-related pulmonary complications such as acute respiratory distress syndrome, pneumonitis, pulmonary embolism, and respiratory failure; COVID-19-related cardiovascular complications such as cardiac arrhythmia, cerebrovascular accident, congestive heart failure, deep venous thrombosis, and myocardial infarction; and other COVID-19-related complications such as acute hepatic injury, bowel perforation, peritonitis, acute renal failure, encephalopathy, and seizures. ^b^ The referent group for COVID-19-related deaths are those that are still living immediately following SARS-CoV-2 infection (T_0_). Deaths due to cancer, complications of cancer treatment, and other or unknown causes were excluded from the categorization of this binary outcome.

**Table 4 cancers-17-00857-t004:** Association of socially constructed race and ethnicity with COVID-19-related outcomes among cancer patients in the ASCO Registry from time of confirmed SARS-CoV-2 infection to end of the acute phase of infection.

Race and Ethnicity	FROM CONFIRMED SARS-CoV-2 INFECTION TO THE END OF THE ACUTE PHASE OF INFECTION ^†^							
	COVID-19-Related Hospitalization		COVID-19-Related Receipt of Supplemental Oxygen		Any COVID-19-Related Complications ^a^		COVID-19-Related Death ^b^	
	RR (95% CI), *p*		RR (95% CI), *p*		RR (95% CI), *p*		RR (95% CI), *p*	
	Model 1 ^††^	Model 2 ^‡^	Model 1 ^††^	Model 2 ^‡^	Model 1 ^††^	Model 2 ^‡^	Model 1 ^††^	Model 2 ^‡^
N	N = 1799	N = 1659	N = 2682	N = 2489	N = 2270	N = 2103	N = 3234	N = 3000
AAPI	1.03 (0.72–1.48), 0.854	1.10 (0.77–1.57), 0.609	0.86 (0.59–1.25), 0.419	0.95 (0.65–1.38), 0.773	1.12 (0.91–1.36), 0.284	1.17 (0.96–1.42), 0.129	0.78 (0.25–2.44), 0.670	0.81 (0.26–2.57), 0.724
AIAN	0.99 (0.69–1.40), 0.941	0.94 (0.62–1.40), 0.749	0.86 (0.58–1.27), 0.445	0.80 (0.51–1.25), 0.320	1.00 (0.78–1.30), 0.973	1.02 (0.77–1.35), 0.909	1.33 (0.52–3.39), 0.552	1.39 (0.56–3.48), 0.482
Hispanic	1.06 (0.84–1.33), 0.635	1.17 (0.92–1.48), 0.210	1.04 (0.85–1.26), 0.718	1.09 (0.89–1.34), 0.400	**1.28** **(1.13–1.45),** **<0.001**	**1.29** **(1.13–1.47),** **<0.001**	1.05 (0.58–1.89), 0.875	1.13 (0.59–2.17), 0.704
NHB	1.19 (0.99–1.43), 0.067	1.21 (0.99–1.47), 0.066	0.92 (0.77–1.09), 0.339	0.97 (0.81–1.16), 0.724	1.09 (0.97–1.22), 0.144	1.10 (0.97–1.25), 0.132	1.00 (0.57–1.75), 0.990	1.02 (0.53–1.95), 0.955
NHW	1.00 (ref.)	1.00 (ref.)	1.00 (ref.)	1.00 (ref.)	1.00 (ref.)	1.00 (ref.)	1.00 (ref.)	1.00 (ref.)

Abbreviations: AAPI, non-Hispanic Asian American and Pacific Islander; AIAN, non-Hispanic American Indian or Alaska Native; NHB, non-Hispanic Black; NHW, non-Hispanic White; *p*, *p*-value; RR, Risk Ratio. ^†^ Independent relative risk regression models were run using a Poisson distribution, log link, and robust error variances to obtain RRs that examined the association of race and ethnicity with COVID-19-related hospitalization, COVID-19-related receipt of supplemental oxygen, any COVID-19-related complications, and COVID-19-related death among cancer patients from time of SARS-CoV-2 Infection to the end of the acute phase of infection (T_0_ − T_1_). The Benjamini–Hochberg (BH) approach was employed to correct for multiple comparisons, using a False Discovery Rate (FDR) of 0.05. Following correction, *p*-values ≤ 0.003 were statistically significant—each significant RR and *p*-value following correction is bolded. ^††^ Model 1 adjusts patient-level covariates–that is, sociodemographic, COVID-19, health history, and cancer status factors measured at time of confirmed SARS-CoV-2 infection. Sociodemographic factors include sex and age at COVID-19 diagnosis; COVID-19 factors include time of COVID-19 diagnosis, vaccination status, and receipt of anti-COVID-19 drugs; health history factors include body mass index, history of tobacco use, and number of comorbidities requiring active treatment in the past 12 months; and cancer status factors include cancer type, cancer stage, receipt of systemic anti-cancer therapy, and last known performance status (ECOG). ^‡^ Model 2 adjusts for all patient-level covariates in Model 1 in addition to area-level measures estimated in the patient’s primary area of residence at the time of cancer diagnosis: census region, percentage of population with only a high school diploma, and median household income. ^a^ Any COVID-19-related complications include the following: COVID-19-related pneumonia, COVID-19-related systemic complications such as bleeding, disseminated intravascular coagulation, sepsis, and multiorgan failure; COVID-19-related pulmonary complications such as acute respiratory distress syndrome, pneumonitis, pulmonary embolism, and respiratory failure; COVID-19-related cardiovascular complications such as cardiac arrhythmia, cerebrovascular accident, congestive heart failure, deep venous thrombosis, and myocardial infarction; and other COVID-19-related complications such as acute hepatic injury, bowel perforation, peritonitis, acute renal failure, encephalopathy, and seizures. ^b^ The referent group for COVID-19-related deaths are those that are still living following SARS-CoV-2 infection by the end of short-term follow-up (T_1_). Deaths due to cancer, complications of cancer treatment, and other or unknown causes were excluded from the categorization of this binary outcome.

**Table 5 cancers-17-00857-t005:** Associations between area-level measures and COVID-19-related outcomes among cancer patients in the ASCO Registry from confirmed SARS-CoV-2 infection to the end of the acute phase of infection.

	FROM CONFIRMED SARS-CoV-2 INFECTION TO THE END OF THE ACUTE PHASE OF INFECTION ^†^							
	COVID-19-Related Hospitalization		COVID-19-Related Receipt of Supplemental Oxygen		Any COVID-19-Related Complications ^b^		COVID-19-Related Death ^c^	
	RR (95% CI), *p*		RR (95% CI), *p*		RR (95% CI), *p*		RR (95% CI), *p*	
	T_0_	T_0_ − T_1_	T_0_	T_0_ − T_1_	T_0_	T_0_ − T_1_	T_0_	T_0_ − T_1_
N	N = 3637	N = 1659	N = 3487	N = 2489	N = 3245	N = 2103	N = 4553	N = 3000
Percentage of population with only a high school diploma ^a^								
Q1	1.00 (ref.)	1.00 (ref.)	1.00 (ref.)	1.00 (ref.)	1.00 (ref.)	1.00 (ref.)	1.00 (ref.)	1.00 (ref.)
Q2	1.07 (0.97–1.18), 0.159	1.12 (0.91–1.37), 0.286	1.16 (1.02–1.32), 0.025	1.07 (0.90–1.28), 0.434	1.07 (0.97–1.17), 0.186	1.10 (0.97–1.24), 0.130	1.13 (0.80–1.61), 0.490	0.89 (0.52–1.54), 0.684
Q3	1.17 (1.05–1.30), 0.005	1.04 (0.81–1.33), 0.762	1.24 (1.06–1.44), 0.006	1.11 (0.90–1.36), 0.323	1.15 (1.03–1.28), 0.011	1.10 (0.95–1.27), 0.189	1.56 (1.05–2.31), 0.029	0.61 (0.31–1.22), 0.163
Q4	1.17 (1.00–1.36), 0.044	1.09 (0.76–1.56), 0.632	1.29 (1.06–1.57), 0.011	1.25 (0.94–1.67), 0.123	1.07 (0.92–1.26), 0.377	1.23 (1.01–1.50), 0.036	**2.12** **(1.33–3.39),** **0.002**	0.71 (0.30–1.71), 0.450
Median household income ^a^								
Q1	0.92 (0.81–1.04), 0.161	0.80 (0.60–1.05), 0.106	0.80 (0.68–0.96), 0.013	0.80 (0.63–1.01), 0.057	0.90 (0.80–1.02), 0.091	0.84 (0.72–0.99), 0.033	0.54 (0.34–0.86), 0.009	1.63 (0.70–3.77), 0.254
Q2	0.96 (0.86–1.07), 0.429	1.08 (0.86–1.37), 0.491	0.94 (0.81–1.08), 0.369	0.99 (0.82–1.21), 0.940	0.99 (0.88–1.10), 0.824	0.95 (0.82–1.09), 0.431	0.75 (0.52–1.08), 0.123	1.96 (1.03–3.71), 0.039
Q3	0.94 (0.85–1.04), 0.222	0.94 (0.75–1.17), 0.557	0.89 (0.78–1.03), 0.109	0.93 (0.78–1.12), 0.443	0.97 (0.87–1.07), 0.502	0.93 (0.82–1.06), 0.290	0.62 (0.43–0.88), 0.009	1.71 (0.93–3.16), 0.087
Q4	1.00 (ref.)	1.00 (ref.)	1.00 (ref.)	1.00 (ref.)	1.00 (ref.)	1.00 (ref.)	1.00 (ref.)	1.00 (ref.)

Abbreviations: AAPI, non-Hispanic Asian American and Pacific Islander; AIAN, non-Hispanic American Indian or Alaska Native; NHB, non-Hispanic Black; NHW, non-Hispanic White; *p*, *p*-value; RR, Risk Ratio. ^†^ Independent relative risk regression models were run using a Poisson distribution, log link, and robust error variances to obtain RRs that examined the association of an area-level social determinant of health factor with COVID-19-related hospitalization, COVID-19-related receipt of supplemental oxygen, any COVID-19-related complications, and COVID-19-related death among cancer patients, separately for time of SARS-CoV-2 infection (T_0_) and time from SARS-CoV-2 infection to the end of the acute phase of infection (T_0_ − T_1_). Each model adjusted for patient-level covariates–that is, sociodemographic, COVID-19, health history, and cancer status factors measured at time of confirmed SARS-CoV-2 infection. Sociodemographic factors include race/ethnicity, sex and age at COVID-19 diagnosis; COVID-19 factors include time of COVID-19 diagnosis, vaccination status, and receipt of anti-COVID-19 drugs; health history factors include body mass index, history of tobacco use, and number of comorbidities requiring active treatment in the past 12 months; and cancer status factors include cancer type, cancer stage, receipt of systemic anti-cancer therapy, and last known performance status (ECOG). Also, the census region of the patient’s primary residence, measured at the time of cancer diagnosis, was adjusted for as an area-level covariate. The Benjamini–Hochberg (BH) approach was employed to correct for multiple comparisons, using a False Discovery Rate (FDR) of 0.05. Following correction, *p*-values ≤0.003 were statistically significant—each significant RR and *p*-value following correction is bolded. ^a^ Each social determinant of health area-level factor was estimated based on census-level data for the patient’s primary area of residence at cancer diagnosis. To prevent reidentification of Registry patients by way of their residential area, this variable was segmented into quartiles. For median household income, the quartiles were as follows: Q1, <USD 43,125; Q2, USD 43,125–USD 54,047; Q3, USD 54,048–USD 68,446; Q4, ≥USD 68,446. For percentage of population with only a high school diploma, the quartiles were as follows: Q1, <25.5%; Q2, 25.5–33.7%; Q3, 33.8–41.2%; Q4, ≥42.2%. ^b^ At time of SARS-CoV-2 infection (T_0_), any COVID-19-related complications include the following: COVID-19-related pneumonia, COVID-19-related systemic complications such as bleeding, disseminated intravascular coagulation, and sepsis; COVID-19-related pulmonary complications such as acute respiratory distress syndrome, pneumonitis, pulmonary embolism, and respiratory failure; COVID-19-related cardiovascular complications such as cardiac arrhythmia, cerebrovascular accident, congestive heart failure, deep venous thrombosis, and myocardial infarction; and other COVID-19-related complications such as acute hepatic injury, bowel perforation, peritonitis, acute renal failure, encephalopathy, and seizures. From time of SARS-CoV-2 infection to the end of short-term follow-up (T_0_ − T_1_), systemic complications additionally include multiorgan failure. ^c^ The referent group for COVID-19-related deaths are those that are still living immediately following SARS-CoV-2 infection (T_0_) or by the end of short-term follow-up (T_1_). Deaths due to cancer, complications of cancer treatment, and other or unknown causes were excluded from the categorization of this binary outcome.

## Data Availability

The authors do not own the rights to the ASCO Registry. However, the data may be available to others upon request from the American Society of Clinical Oncology Inc.

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
