# Peer review of "Differences in COVID-19-Related Hospitalization, Treatment, Complications, and Death by Race and Ethnicity and Area-Level Measures Among Individuals with Cancer in the ASCO Registry"

_cancers, 2025, doi:10.3390/cancers17050857_

Round 1
Reviewer 1 Report
Comments and Suggestions for Authors
We congratulate the authors on the effort put on the present analysis which provides very valuable information. The amount of information is significant and complex but extensively and adequately reported, and the discussion is sound.
I have one minor comment for your consideration:
-The authors have described complications, treatments etc a two different timepoints: and 'early' one, “at the time of a positive COVID-19 test”/“at the time of positive/confirmed SARS-CoV2-infection” and a 'late' one at 1-3 months. Therefore I understand that the ‘early’ period comprises complications that take place the first month after a confirmed diagnosis, meaning the acute phase of the infection, and not a transversal time point. Consider reprhasing.
Reviewer 2 Report
Comments and Suggestions for Authors
In this manuscript, the authors conducted one retrospective study related to differences in COVID-19-related hospitalization, treatment, complications, and death by race and ethnicity and area-level measures among individuals with cancer in the ASCO registry. They found that cancer patients from racial and ethnic minority groups, as well as those living in socioeconomically disadvantaged areas, have a significantly greater risk of poorer COVID-19-related outcomes than their NHW counterparts and those living in less disadvantaged areas. The analysis of this study was not adequate. Below are a number of issues that the authors shall address or revise:
- Some important results should be summarized in figures and more analysis should be applied to support the conclusions. For example, the ROC curve can be used to study the relationship of COVID-19-related prevalence.
- The cancer types can also be attributed differently in the results. The authors can analyze different types of cancers, especially lung cancer.
- In the conclusion part, the authors just listed the factors mentioned in this study. The authors should give some perspectives or hypotheses on the reasons caused these phenomena and give some suggestions to reduce the differences.
Reviewer 3 Report
Comments and Suggestions for Authors
Dear Authors,
I would like to congratulate you on a well-structured and insightful manuscript. The study presents a rigorous analysis of the effects of COVID-19 on individuals with cancer and provides a substantial contribution to the existing body of knowledge. The thorough evidence provides a valuable foundation for future analyses of this important topic.
I particularly appreciate the comprehensive analytical approach and the clear presentation of the results in the tables. I am sure that the logical structure and coherent narrative will allow the readers to easily follow the article.
Given the high quality of the manuscript and its potential impact on further research, I recommend it for publication without further revision. Thank you for the opportunity to review this excellent paper.
Sincerely,
Reviewer 4 Report
Comments and Suggestions for Authors
People with cancer who were exposed to SARS-CoV-2, the virus that causes COVID-19, were more likely to develop COVID-19 complications and die than people without cancer. During the COVID-19 pandemic, racial and ethnic minorities were more likely to contract the virus, be hospitalized, or die than non-Hispanic whites. Therefore, the authors examined COVID-19-related hospitalizations, supplemental oxygen requirements, multi-organ complications, and deaths in a large sample of multiethnic cancer patients with SARS-CoV-2 infection from the American Society of Clinical Oncology COVID-19 Registry. Cancer patients from racial and ethnic minority groups, as well as those living in socioeconomically disadvantaged areas, were found to be at significantly higher risk for worse COVID-19-related outcomes than their non-Hispanic white counterparts and those living in less disadvantaged areas.
I was concerned about confounding factors (gender and age), and the manuscript took care to eliminate these. I can therefore conclude that the methods and results are sound.
The results of the current study also showed that NHB, Hispanic/Latinx, and AIAN cancer patients have an increased risk of developing COVID-19-related complications. Due to the distribution of upstream factors (social disadvantage, risk exposure, and social inequities in education, employment, housing, and access to resources), many racial and ethnic minority individuals were more vulnerable to viral infection and food/supply shortages, less likely to have received vaccinations to help mitigate the severity of COVID-19, and less likely to receive timely, high-quality medical care during the pandemic. All of these factors contribute to an increased risk of more severe acute illness and long-term sequelae following infection with SARS-CoV-2. In addition, these vulnerable groups tend to live disproportionately in social environments and communities with high concentrations of unhealthy food, tobacco, and alcohol (i.e., food deserts, food swamps, areas with high density of alcohol and tobacco retailers), which further increases the risk of developing chronic conditions (diabetes, CVD, obesity) that compromise immunity and contribute to worse COVID-19 outcomes. In this way, suggestions for improving the social environment are also made, which is good.
Of course, the Editor will make the final decision, but as a Reviewer, I hope that this manuscript will be published soon.
Reviewer 5 Report
Comments and Suggestions for Authors
Dear Authors,
Thank you for the opportunity to review your manuscript, which provides valuable insights into the intersection of COVID-19 and cancer-related health disparities. Your study makes an important contribution by highlighting how race, ethnicity, and area-level socioeconomic factors influence COVID-19 outcomes among cancer patients. Using a large, multiethnic dataset from the ASCO Registry, combined with robust statistical analyses, strengthens the credibility and impact of your findings. Additionally, your emphasis on the role of structural and social determinants of health (SDOH) aligns well with current efforts to address healthcare inequities.
I have a few minor suggestions to further refine the manuscript and enhance its clarity and impact.
The introduction and abstract could provide a more explicit justification for the study by better identifying gaps in the existing literature. Clearly outlining what is not yet known about racial and socioeconomic disparities in COVID-19 outcomes among cancer patients would help underscore the novelty of your findings.
In addition, the discussion of race and ethnicity as "socially constructed categories" is a strength, but it would be beneficial to briefly acknowledge potential classification biases in the dataset and how they might influence the results. A short reflection on measurement limitations could add further rigor to the study.
The conclusion effectively emphasizes the importance of addressing healthcare disparities, but it would be even stronger with specific recommendations on how policymakers or clinicians might apply these findings to improve patient care and reduce inequities.
Congratulations on this well-conducted research, and I appreciate the opportunity to review your work.
Round 2
Reviewer 2 Report
Comments and Suggestions for Authors
I am satisfied with the author’s responses to my issues raised in my initial review. The revised manuscript is easier to follow based on feedback from the reviewers. I recommend that the revised paper be accepted.